# Advantages and Disadvantages of Using Magnetic Nanoparticles for the Treatment of Complicated Ocular Disorders

**DOI:** 10.3390/pharmaceutics13081157

**Published:** 2021-07-27

**Authors:** Elena K. Schneider-Futschik, Felisa Reyes-Ortega

**Affiliations:** 1Department of Biochemistry & Pharmacology, Faculty of Medicine, School of Biomedical Sciences, Dentistry and Health Sciences, The University of Melbourne, Parkville, VIC 3010, Australia; elena.schneiderfutschik@unimelb.edu.au; 2Visual Quality Research Group, Department of Ophthalmology, Maimonides Biomedical Research Institute of Cordoba (IMIBIC), Reina Sofía University Hospital and University of Cordoba, 14004 Cordoba, Spain

**Keywords:** ocular disorders, ocular cancer, cystic fibrosis, magnetic nanoparticles, therapeutic agents, magnetic diagnosis agents

## Abstract

Nanomaterials provide enormous opportunities to overcome the limitations of conventional ocular delivery systems, such as low therapeutic efficacy, side effects due to the systemic exposure, or invasive surgery. Apart from the more common ocular disorders, there are some genetic diseases, such as cystic fibrosis, that develop ocular disorders as secondary effects as long as the disease progresses. These patients are more difficult to be pharmacologically treated using conventional drug routes (topically, systemic), since specific pharmacological formulations can be incompatible, display increased toxicity, or their therapeutic efficacy decreases with the administration of different kind of chemical molecules. Magnetic nanoparticles can be used as potent drug carriers and magnetic hyperthermia agents due to their response to an external magnetic field. Drugs can be concentrated in the target point, limiting the damage to other tissues. The other advantage of these magnetic nanoparticles is that they can act as magnetic resonance imaging agents, allowing the detection of the exact location of the disease. However, there are some drawbacks related to their use in drug delivery, such as the limitation to maintain efficacy in the target organ once the magnetic field is removed from outside. Another disadvantage is the difficulty in maintaining the therapeutic action in three dimensions inside the human body. This review summarizes all the application possibilities related to magnetic nanoparticles in ocular diseases.

## 1. Introduction

The anatomy of the human eye can be divided in two main sections: the anterior segment of the eye, which is formed by the cornea, conjunctiva, aqueous humor, iris, ciliary body, and lens, and the posterior segment of the eye, which mainly consists of the vitreous humor, retina, choroid, and optic nerve. The most common diseases that affect the anterior segment of the eye are dry eye syndrome, conjunctivitis, anterior uveitis, cataract, and keratoconus [1]. The most prominent diseases affecting the posterior segment of the eye are age-related macular degeneration, diabetic retinopathy macular edema, proliferative vitreoretinopathy, posterior uveitis, and cytomegalovirus [2]. Furthermore, glaucoma is a very common disease that can affect both sections of the eye.

Drug administration is required for the treatment of all these diseases and treatment can vary from less invasive to highly invasive: topically, systemic injection, subconjunctival, intracameral, and intravitreal injection. The efficacy of the administrated drug depends on its administration route, since each section and subsection display different anatomical, physicochemical, or physiological barriers [3]. Physicochemical factors, such as the solubility, molecular weight, and size of drugs, considerably affect drug absorption and, hence, therapeutic efficacy in the organism. Unionized drugs easily permeate biological membranes. Conversely, ionized molecules, especially anionic compounds, tends to be retained or repelled by the corneal epithelium since this layer shows a negative charge at physiological pH. Drugs with a high molecular weight (Mw > 5000 Da) are difficult to penetrate the corneal epithelium, the conjunctiva, and the sclera [4]. The most restricted layer for high molecular weight drugs is the corneal epithelium due to the tight junctions that it presents (less than 2 nm diameter pores).

Cystic fibrosis (CF) is an autosomal recessive genetic disorder that affects primarily the lungs and also the pancreas, liver, and intestine [5,6]. This is the most common and severe genetic disease in Caucasian population and show a high morbidity and mortality due to the abnormalities created in the pulmonary and gastrointestinal tract. From 1994 to 2010, there were 5130 deaths (2443 in males and 2687 in females) identified in Europe alone, all between the ages of 0 and 30 years [7]. The highest rates in Europe are in the United Kingdom and Ireland, with 1.37 and 2.98 per 10,000 inhabitants [8], respectively. This autosomal recessive disease is caused by a defective membrane protein known as cystic fibrosis transmembrane conductance regulator (CFTR), which causes an abnormal transport of chloride/sodium and water across the epithelial surfaces in the gastrointestinal and respiratory tracts, the reproductive system, and the sweat glands due to the thick and viscous secretions that it forms [9,10]. To date, more than 1000 mutations of the CF gene are known, depending on which aspect the mutation of the CFTR has an effect (function and/or synthesis) [11,12].

CF patients overexpress a number of different cytokines through their impaired immune response system [13]. Several of these cytokines are involved in the regulation of the tear film and they can promote ocular inflammation [14]. The CFTR protein functions as chloride channel on the apical membrane of epithelial cells and is responsible for the regulation of the secretion of chloride ions and the re-absorption of sodium ions. Cl^−^ flux is needed for the maintenance of various ocular structures, including the corneal epithelium, corneal endothelium, conjunctival epithelium, and retinal pigment epithelium. Therefore, the effects on CFTR malfunction (mutations interfering with protein synthesis, protein maturation, mutations altering channel regulation, and alterations on chloride conductance) can directly cause alterations to all these ocular structures.

CF disease limits the airflow in the lungs (hypoxia), which is often associated with an inflammatory response and abnormal pulmonary function of the lungs. This inflammatory response can be detected at the very early stage of the disease [15]. Furthermore, this status of hypoxia can also affect the optic nerve and retinal cells and retinal nerve fibers, altering the visual field. Additionally, the chloride ion channel alteration by the CFTR can also affect the ocular physiology. The ion Cl^−^ is the major component of the corneal wound electric current: it contributes to the basal tear production in the corneal and conjunctival epithelium, facilitates the preservation of the corneal transparency, and contributes to the regulation of the subretinal space and normal retinal pigment epithelium [16]. Castagna et al. demonstrated, in a clinical study carried out with 40 CF patients who suffered from reduced tear secretion, conjunctival abnormalities and reduced lens transparency [17]. The underlying cause for this was related to the digestive insufficiency and indirectly to the subsequent low plasma concentration of vitamin A [16]. Another clinical study carried out by Seliger et al. demonstrated that 91.1% of CF patients were diagnosed with cataract disease [18], which is in line with another clinical study conducted in the US evaluating the risk of cataracts of pediatric patients with cystic fibrosis [19]. More aggressive ocular problems were described by Starr et al. in a 35-year-old male with CF that presented intraretinal hemorrhages along the nerve fiber layer with associated retinal thickening in the inferior macula, which means an acute branch retinal vein occlusion in his left eye with associated macular edema [20]. Rottner et al. showed that CF patients are at high risk of developing retinal vein occlusions due to several systemic thrombogenic factors, such as diabetes, hypertension, hyperlipidemia, an elevated level of fibrinogen, hyperhomocysteinemia, or hypergammaglobulinemia [21]. A pilot study carried out with 11 patients (6 male and 5 female) by Nebbioso et al. suggests that CF disease causes malfunction of the magnocellular system in early glaucoma or diabetic retinopathy patients [22], probably due to the oxygen alterations of the ganglion cells.

Nanotechnology approaches have brought about a revolution in the drug administration field, increasing the bioavailability of drugs, decreasing the toxicity and secondary effects, and allowing for targeted therapy. In general, nanosize devices are suitable to provide sustained drug delivery and gene therapy thanks to perfect control of the surface-area-to-volume ratios that improve tissue penetration [3]. The drug nanocarriers most used in ocular delivery applications are nanosuspensions, liposomes, dendrimers, vesicles, niosomes, nanospheres, nanomicelles, and nanoparticles [23].

Magnetic nanoparticles are a kind of nanoparticle that when in the presence of an external magnetic field, their magnetic spins tend to align in the same direction of the field, resulting in an induced magnetization that allows their use in the magnetic resonance imaging diagnosis technique [24]. Besides, they can act as hyperthermia agents (cancer therapy) [25], can increase the release rate of different drugs (magnetically triggered drug release) [26,27], and are able to facilitate the accumulation in certain tumor areas due to their magnetofection (target therapy) [28]. These kinds of nanoparticles are very useful for the diagnosis and treatment of different tumors. In the ophthalmology field, it is a challenge to get an efficient therapeutic drug delivery system due to the anatomical, physicochemical, or physiological barriers. These complications are even more challenging in the case of ocular tumors. It has been demonstrated that biocompatible magnetic fluids or magnetic nanoparticles can be directly administrated into the ocular tumors [29]. Once there, the magnetic suspension can be placed under an alternating magnetic field with a frequency between 100 and 500 kHz (safe frequency for humans), generating a localized heating via the mechanisms of hysteresis and relaxation losses [30]. Increasing temperatures can facilitate perfusion within the tumor and higher chemotherapeutic drug delivery [29,31]. Magnetic nanoparticles are suitable to be loaded with anticancer drugs acting as drug carriers able to control and modulate drugs release thanks to their response to a magnetic field [27]. This review is focused in the applications of nanoparticles that possess magnetic properties, highlighting their advantages as therapeutic ocular agents and their limitations.

## 2. Ocular Barriers for Drug Delivery Systems

Initially, the anatomical barrier that a drug finds in the eye is the cornea (Figure 1). Cornea thickness is from 551 to 565 µm in the center section and from 612 to 640 µm in the periphery, and is formed by five layers: epithelium, Bowman’s layer, stroma, Descemet membrane, and the endothelium [32]. The cornea contains barriers to both hydrophilic and hydrophobic molecules. Corneal epithelium is the first anatomical barrier, which consists of a basal layer of columnar cells surrounded by intercellular tight junctions [33]. Corneal epithelium is negatively charged at physiological pH and the tight junctions act as barriers for the permeation of hydrophilic drugs, such as fluoroquinolones (e.g., norfloxacin) or intraocular pressure (IOP) beta-blockers (e.g., timolol). After the epithelium, the second cornea layer that can be an impediment for drug diffusion is the stroma (Figure 1), which is formed by multiple layers of hexagonally arranged collagen fibers containing aqueous pores that allow hydrophilic drugs to easily pass through but acting as a significant barrier for lipophilic drugs [4], such as prostaglandin (e.g., latanaprost) or NSAIDs (e.g., celecoxib). Consequently, drug bioavailability depends of the balance between lipophilicity and hydrophilicity to avoid being retained in the corneal barriers. Recently, it has been observed that the use of MNPs coated with chitosan can enhance the corneal residence time and drug contact [34], since the chitosan coating acts as a permeability enhancer.

After the cornea, the next anatomical barrier is the iris and ciliary body (Figure 1). The iris is composed of a pigmented stromal layer formed by epithelial cells. The ciliary body is formed by ciliary muscles and ciliary epithelial cells that produce aqueous humor, the transparent protein-containing fluid that nourishes the cornea and lens [35]. The constant secretion, flow, and drainage of aqueous humor controls the IOP of the eye. Then, MNPs that are able to penetrate the cornea need to diffuse against the flow of the aqueous humor and face elimination via Schlemm’s canal [35]. Another anatomical barrier is the lens that is formed by four structures: lens capsule, epithelium, cortex and nucleus. The epithelial cells that formed the capsule generates a barrier to hydrophilic molecules. The cortex and nucleus are made by lens fibers, whose tightly compact arrangement limit drug diffusion in the lens [36]. Conjunctiva and sclera are other anatomical barriers of the eyes, whose composition is basically collagenous and elastic fibers, limiting, such as in the cornea, the drug diffusion [37] (Figure 1). The conjunctiva runs 3–5 cell layers thick, with tight junctions on the apical surface. The sclera is a fibrous, opaque tissue forming the outer layer of the eye and is continuous with the cornea. Choroid is a layer of vasculature lying between the retina and sclera and it contains Bruch’s membrane, which is formed by several layers of collagenous and elastic fibers and forms the basement membrane for retinal pigment epithelium. Due to the systemic circulation of this layer, there is a rapid clearance of the drug delivery systems [38].

The retina is anatomically the back layer of the eye and is composed of the blood–retinal barrier (BRB) (Figure 1). The inner part of the BRB is formed by tight junctions between the retinal capillary endothelial cells, and the outer-blood–retinal barrier separates the choroid and Bruch’s membrane from the inner retina. Drug delivery systems must be able to cross the inner and outer membranes of the retina, which pose a strict physical barrier for most polymeric nanoparticles. Magnetic nanoparticles can enhance the retina layers penetration due to their magnetic response under the application of an external magnetic field [39,40].

The main physiological barriers of the eyes are its tear film and nasolacrimal drainage. The lacrimal fluid is an isotonic aqueous solution that contains a mixture of proteins and lipids. The lacrimation and nasolacrimal drainage decrease the drug exposure time during their administration, and they are responsible for a significant loss of topically applied drugs [41]. Another physiological barrier is related to the efflux proteins, which are located on the apical or basolateral cell membranes of the corneal epithelium, in conjunctiva, and in the iris ciliary body [42]. These proteins can also affect to the absorption of different drugs. Mainly there are two efflux pumps responsible for drug resistance: the P-glycoprotein that limit the entry of amphipathic drugs, and the multidrug-resistant protein (MRP), which acts as an organic anionic transporter [43].

## 3. Ocular Delivery Routes and Their Limitations

Focusing in the biodistribution of the magnetic nanoparticles (MNPs) administrated to the eye, several factors have to be taken into account during MNPs synthesis, such as the magnetic surface properties, size, suspension media, and administration route. Different drug delivery routes can be used to inject nanoparticles into the eyes, such as topical, systemic, punctal, subconjunctival, intrascleral, fornix, sub-Tenon’s, suprachoroidal, subretinal, and intravitreal injection [44] (Figure 2).

Topical application has the limitation of reaching a high ocular bioavailability as a large portion of the compound applied would be lost due to dilution of tears and lacrimation. Usually, the amount of drug that reaches the aqueous humor is less than 5% (Figure 3). Sustained release of the drug is difficult and forces one to markedly increase the drug dose administrated in order to get an optimum therapeutic efficacy.

Systemic injection also has the same limitation of a low amount of drug available in the specific site (the eye), especially if the drug is more hydrophilic. It would require more frequent systemic injections to reach and maintain a therapeutic dose, which creates more side effects in the body. The periocular injection can refer to posterior juxtascleral, subconjunctival, retrobulbar, peribulbar, or subtenon injection. Some risks associated for periocular injections are hyphemia, an increase in intraocular pressure, corneal decompensation, and even strabismus. Intravitreal injections are becoming a more popular choice for ocular drug delivery. By micro-needle injection of the compound directly into the vitreous, intravitreal injection could offer a higher drug load in the retina and vitreous compared to other delivery methods. Nowadays, there are some intravitreal products in the market, such as Avastin^®^ (Genentech, San Francisco, CA, USA), Lucentis^®^ (Genentech, San Francisco, CA, USA), Macugen^®^ (Eyetech Inc., Boca Raton FL, USA), and Triesence^®^(Alcon, Vernier-Geneva, Switzerland) [45]. The drug’s molecular weight is a major factor affecting drug elimination for intravitreal injection. The disadvantages of intravitreal injection include development of certain complications, such as intravitreal hemorrhages, endophthalmitis, and retinal detachment. Patients with diseases affecting the posterior segment usually need multiple intravitreal injections and follow careful monitoring [46].

## 4. Ocular Disorders Derived from Cystic Fibrosis Disease

The corneal epithelium contributes to fluid transport via various ion channels, from the stroma to the pre-corneal tear film. Numerous channels in the corneal epithelium are responsible of the fluid transport in the eye (Figure 4). Located on the basolateral membrane, the sodium:potassium:chloride (Na^+^:K^+^:2Cl^−^) co-transporter functions in parallel with the sodium:potassium (Na^+^:K^+^) pump resulting in CF influx. The rate of secretion by these channels is regulated by the CF conductance of the apical membrane. In addition to CFTR, calcium-activated CF channels (CLCA2) are also thought to contribute to CF efflux. At the ocular surface, basal CFTR activity is likely minimal, as CF patients with loss-of-function CFTR mutations suffer only from mild tear film abnormalities [47,48,49,50]. However, the CFTR-facilitated Cl^−^ transport at the ocular surface provides a rational basis for the investigation of CFTR modulators, e.g., ivacaftor. Currently, eye manifestations of CF are less well known; however, mounting evidence suggests that ocular disorders in CF are a serious problem. A number of ocular disorders derived from CF have been reported, including xerophthalmia, papilledema, and retinal hemorrhages [51]. Joshi et al. [52] reported the first case of newly diagnosed CF-related liver disease in a teenage boy presenting symptoms of night blindness secondary to vitamin A deficiency. In CF, the malabsorption of fat-soluble vitamins, e.g., vitamin A, reduces the concentration of the retinol-binding protein that is essential for the liver-to-tissue transport of retinol. Night blindness is the first sign of vitamin A deficiency with further symptoms after prolonged periods of deficiency, such as Bitot’s spots; triangular, perilimbal grey plaques of keratinized conjunctival debris; and xerosis and dry granular patches. Furthermore, patients with CF have been reported to develop retinal vein occlusions [20,53,54]. It has been hypothesized that elevated fibrinogen levels due to chronic infections or increased homocysteine levels predispose patients with CF to develop retinal vein occlusions.

Other ocular disorders related to CF disease are differences in the morphology of the cornea, a reduction in the endothelial cell area, an increase in corneal thickness, an increase in the endothelial cell density and permeability, and an increase in the endothelial pump rate. Lass et al. [55] observed morphological differences in the corneal endothelium in CF patients and CF-related diabetes patients. Mean corneal thickness was significantly greater for both CF groups compared to a safe patient and, therefore, the corneal endothelial permeability and mean relative pump rate were significantly higher in the two CF groups. The increased corneal thickness observed in the CF group suggests there is only partial compensation by the increased pump rate for the increased permeability. Several studies have investigated the incidence of blepharitis in patients with CF [56]. Mrugacz et al. [57] suggested increased blepharitis could indicate lipid dysfunction in CF and that meibomian dysfunction was consistent with the glandular dysfunction observed in CF. Conjunctival xerosis is characterized by keratinization and drying of the conjunctiva due to loss of goblet cells and basal cell proliferation. This incidence of conjunctival xerosis in CF has been reported by several authors [58,59,60]. Macular pigment is derived from two carotenoids, lutein and zeaxanthin. Both the serum lutein and zeaxanthin concentrations as well as the macular pigment optical density were observed to be significantly lower in CF patients [61]. Even at a very early stage of CF disease development (newborns), ocular disorders have been observed [62,63]. As babies with CF frequently present with lower birth weights (which itself has been associated as a risk factor for ametropia, strabismus, and amblyopia), early and regular eye examinations for all children with CF remain essential.

## 5. MNPs as Carriers for Drug Delivery

Magnetic nanoparticles differ from the rest of the nanocarriers due to their magnetic properties that make them unique for drug delivery. Drugs molecules can be conjugated to the shell of magnetic nanoparticles to be injected into the body and be concentrated in a local area (avoiding the damage to other tissues) due to the effect of an external magnetic field. Owing to the MNPs large surface-to-volume ratio, it offers numerous chemically active sites for biomolecule conjugation [64]. It helps to increase the drug circulation time into the organism and to get the target site. Furthermore, these functionalized magnetic nanoparticles can act as hyperthermia agents, providing a more potent therapeutic effect since the increase in temperature in a specific site promote tumor cell death without altering normal cells [65,66]. Besides, magnetic nanoparticles can be easily visualized by magnetic resonance imaging (useful for diagnosis) by the application of an external magnetic field [67]. The most frequent magnetic nanoparticles used in biomedical applications are magnetite (Fe_3_O_4_) or maghemite (ɣFe_2_O_3_) due to their higher biocompatibility and stability [68]. The physicochemical properties of MNPs (such as size, shape, charge, and anisotropy) also are known to affect cellular responses, such as the internalization rates and mechanisms or cytotoxicity [69,70]. It is crucial to predict all these parameters in the nanoparticles synthesis to get the maximum therapeutic treatment. Iron oxide nanoparticles (IONPs) have been described to be easily modulated in terms of size, shape, and magnetic hyperthermia response [65] (Figure 5). In order to corroborate the chemical structure of the IONPs, Fourier transform infrared spectroscopy (FTIR) is often used to characterize the proper nanoparticles formation. As an example, Figure 6 shows a FTIR spectrum of magnetite spherical nanoparticles. It is observed the characteristic bands of magnetite at around 590 cm^−1^ and 400 cm^−1^ correspond to the Fe-O/Fe-O-Fe bonds of magnetite.

The most-often used coated polymers for MNPs are dextran [71], chitosan [72], poly(ethylenglycol) (PEG) [73], poly(lactic-co-glycolic acid) (PLGA) [74], poly(caprolactone) (PCL) [75], silicone [76], and liposomes [77]. These coatings help to stabilize the iron oxide core and avoid aggregation, but they usually decrease the magnetization saturation of bare iron oxide nanoparticles. All these material coatings have been already approved by the FDA to be used as nanoparticle-based medicines in clinical trials [78]. Translating the magnetic nanocarriers to ocular applications include taking into account the interaction that can occur between the functional groups of the coating materials and the collagen of the corneal stroma, since it is considered the major resistance factor during the drug penetration process [3]. It has been demonstrated that PEG and its derives are the most efficient coatings in magnetic nanoparticles in terms of avoiding the ocular physiological barriers [36].

## 6. Current Uses of MNPs as Pharmaceutical Formulations: Ocular Cancer Diagnosis and Treatment

### 6.1. Ocular Diagnosis Techniques

The most common clinical ocular diagnosis techniques are optical coherence tomography (OCT), fluorescein angiography, fundus photography, positron emission tomography (PET), magnetic resonance imaging (MRI), ultrasonography, and confocal microscopy. All of these techniques can be improved by increasing either the sensitivity or resolution using magnetic nanoparticles. For instance, using the ultrasound technique provides cost-effective and real-time imaging; however, it requires the use of contrast agents in the micrometer size range, which are usually unstable saline bubbles that are confined to the vascular system. This can be avoided through the use of magnetic contrast agents that further noticeably increases the resolution of the ultrasound images [79]. OCT combined with magnetic or plasmonic nanoparticles was demonstrated to improve the cross-section absorption about five orders of magnitude larger than conventional indocyanine green in the near-infrared spectral region [80]. The MRI technique is convenient for monitoring the progress of ocular diseases such as diabetic retinopathy, age macular degeneration, ocular tumor angiogenesis, etc. It provides a good spatial resolution but low sensitivity. Using MNPs as a contrast agent in MRI noticeably increases the sensitivity and capability of this technique [81]. The PET technique shows the disadvantage of containing radiolabeled tracer quantities, which can easily be harmful for the patient. The use of MNPs in PET helps to avoid this damage, generating a potentially diagnostic and therapeutic technique very useful in theranostics [82].

The two most important parameters to control in MNPs, to verify their utility in diagnosis techniques, are size and shape. Both variables can affect to the dynamics of the magnetic moments (magnetic saturation, magnetic relaxation); they determine the detection capability as well as affect the internalization and eventual fate of the MNPs inside mammalians [68]. Table 1 summarizes the different MNPs used in clinical trials for the diagnosis of several ocular diseases.

### 6.2. Magnetic Nanoparticles for Ocular Cancer Treatment

MNPs are the most attractive nanomaterials in the treatment of different tumors, including ocular tumors. Several research works described their use in vitro and in vivo as therapeutic agents for the treatment of certain ocular tumors, such as retinoblastoma, uveal melanoma, choroidal melanoma, and choroidal hemangioma. For example, Demirci et al. demonstrated that IONPs can act as efficient therapeutic nanoheaters able to target exclusively the drug delivery in the eye by intravitreal injection and the application of an external magnetic field for the treatment of retinoblastoma [29]. The IONPs used in this study were coated with dextran and were tested in the Y79 retinoblastoma cell line, resulting in selectively killing retinoblastoma tumor cells via the activation of apoptotic pathways. Latorre et al. used gold nanoclusters coated with albumin for the treatment of uveal melanoma [89]. These nanoclusters were loaded with the AZD8055 drug, a selective inhibitor of mTOR that prevents the proliferation of uveal melanoma tumor cells. The therapeutic efficacy was tested in vitro and compared with non-tumoral keratinocytes, showing the high selectivity of the tumor cells. These AZD8055 nanoculsters were tested in vivo, using a mouse model, demonstrating the potential for stopping uveal melanoma metastasis. Giannaccini et al. demonstrated that intravitreally injected MNPs were able to localize rapidly in the retinal pigment epithelium (RPE) as a potent therapeutic tool for the treatment of choroidal melanoma [39,40]. These MNPs were functionalized with the vascular endothelial growth factor to produce transcytosis from the RPE towards more posterior layers in the eye. Orynbayeva et al. studied the internalization of polymeric-coated MNPs on primary rat endothelial cells, showing that MNPs are potent targeting and therapeutic agents that did not affect the structural integrity and functionality of the primary endothelial cells [88]. Yanai et al. used IONPS for intravitreal injection or via the tail vein in a transgenic rat model of the retina, showing the efficacy of targeting the upper hemisphere of the rodent retina [90].

However, there is very little available in the literature regarding the transition from preclinical to clinical studies of magnetic hyperthermia for ocular application. Until now, all published works report on preclinical models in isolation, but none of them talk about how to get from the animal model to humans.

### 6.3. Advantages of the Use of MNPs in Ocular Applications

Magnetic particles have been previously described in the literature as toxic ocular agents due to the iron accumulation that can damage the photoreceptors and interfere with retinal electrophysiology; they also tend to aggregate and can oxidate [91,92]. However, their toxicity highly depends on the particle sizes, the coating used, and the concentration administrated. Raju et al. [93] demonstrated that magnetic microparticles (4 µm mean diameter particles) produced toxicity when they were administrated intravitreally or into the anterior chamber, especially in the corneal endothelium. However, when tested magnetic nanoparticles of 50 nm mean diameter, it was observed to not accumulate in the eye, and therefore no toxicity was found in any layer of the eyes. The same authors demonstrated that the MNPs were safe for intraocular use since they tested the intravitreal and anterior chamber injections of the MNPs into the eye without any signs of toxicity on the retinal structure, photoreceptor function, or aqueous drainage in the eye [94]. In this study, none of the magnetic particles increased the IOP. In fact, the FDA approved the use of several superparamagnetic iron oxide nanoparticles as contrast agents in magnetic resonance imaging (Feridex^®^ Berlex Laboratories, USA; Endorem^®^ Guerbet, France; Sinerem^®^ Guerbet, France; Resovist^®^ Bayer, Germany; Cliavist^®^ Bayer, Germany, and Faraheme^®^ AMAG Pharmaceuticals, Waltham, MA, USA) since they were demonstrated to be safe for humans [95]. Some others are under clinical trials, such as Ferumoxytol [95]. The safest magnetic nanoparticle used are formed by a magnetite (Fe_3_O_4_) or maghemite (γFe_2_O_3_) core and the shell is formed by a biopolymer. Functional groups could be a wide range of molecules, such as carboxyl, antibodies, amines, biotin, and streptavidin, and such functional groups can be attached via disulfide cross-linkers [96,97]. Current pharmacological approaches involve intravitreal injections of a biomedical agent to prevent the aberrant growth of blood vessels. Magnetic nanomaterials provide novel opportunities to reach the back of the eye, thanks to the application of an external magnetic field, and are capable of the encapsulating and modulated-delivery of small molecules. These magnetic nanomaterials can be adopted easily by ocular drug delivery systems to improve current therapies, overcoming the limitations of barriers in vivo and reducing the risk of severe complications that can improve the bioactivity and bioavailability of ocular therapeutic agents.

MNPs have also been studied as potential and useful bacterial detection and bacterial separation agents due to their magnetic properties and antimicrobial effect. Li et al. demonstrated that IONPS under the application of a magnetic field were able to promote antimicrobial effect in biofilm matrixes causing detachment of several bacteria, such as methicillin-resistant *Staphylococcus aureus* (MRSA) [98]. MNPs have also been demonstrated to be good contrast agents for in vivo bacterial imaging due to their superparamagnetic properties. They also improve the antimicrobial efficacy and enhance the bioavailability, reducing systemic side effects [99]. CF patients develop a different bacterial infection that produce a very viscous mucous that is difficult to penetrate by therapeutic molecules. Magnetic hyperthermia can be effectively used to decrease biofilm and mucus viscosity, while enhancing drug and immune cell penetration into the target areas [100]. Moreover, the produced increase in temperature noticeably reduces the formation and growth of biofilms [66]. Thus, magnetic hyperthermia results in increased bacterial membrane permeability, resulting in enhanced targeted killing of bacteria [101].

Patients who suffer from ocular disorders caused by CF disease can be treated with MNPs, as part of a double therapeutic aim: improving the bioavailability of the CF treatment (e.g., using antibiotics and or antimicrobial functionalized MNPs) and treating the ocular disorder with a target-specific drug, reducing the toxicity of both therapeutic agents. The use of a single magnetic nanocarrier can be useful for the treatment of both diseases, as well as improving both therapeutic treatments.

### 6.4. Limitations of the Use of MNPs in Ocular Applications

Some of the limitations of magnetic nanoparticles in drug delivery is that they cannot be concentrated into a three-dimensional space, since the application of an external magnetic field organizes the MNPs into a two-dimensional area. In addition, it is difficult to keep the magnetic particles in the targeted organ once the magnetic field is removed from outside. Another inconvenience is related to the time exposure to the magnetic field: patients cannot be unlimitedly exposed to an external magnetic field, so the therapeutic efficacy is limited to the frequency, intensity, and exposure time of the magnetic field.

### 6.5. Future Perspectives

In the literature, a lack of research studies related to the applications of MNPs for clinical applications is observed. There is still much unknown about the long-term effects of the application of MNPs in medicine. In order to improve the advances in biomedical applications (such as diagnosis, cancer detection, cancer therapy, ocular disease therapy, and respiratory disease therapy), further studies are required to prove the MNPs’ efficacy and safety. More translational studies are required that combines different therapeutic treatments to reduce cost, facilitate the target therapy’s efficacy, and minimize the side effects in patients.

From a global perspective, for the immediate future, targeting strategies in combination with magnetic hyperthermia treatment will be the best initial approach for the majority of ocular diseases, especially those related to CF, for the following three main reasons: less treatment cost, minimally invasive, and higher therapeutic efficiency (less toxic and longer drug activity).

## Figures and Tables

**Figure 1 pharmaceutics-13-01157-f001:**
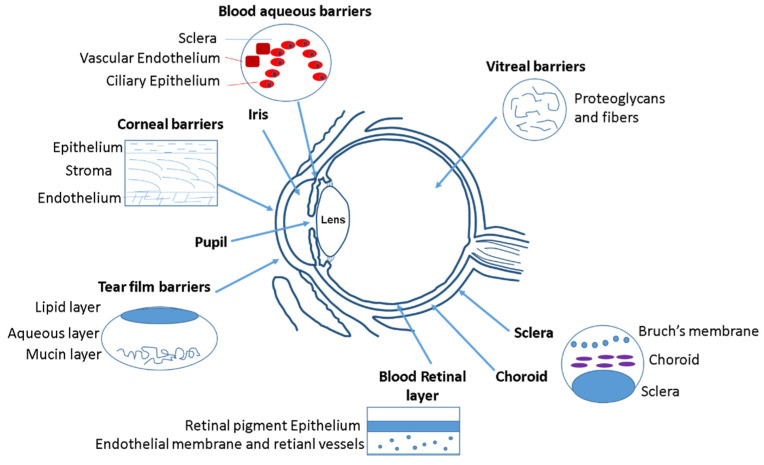
Different ocular barriers for drug delivery systems.

**Figure 2 pharmaceutics-13-01157-f002:**
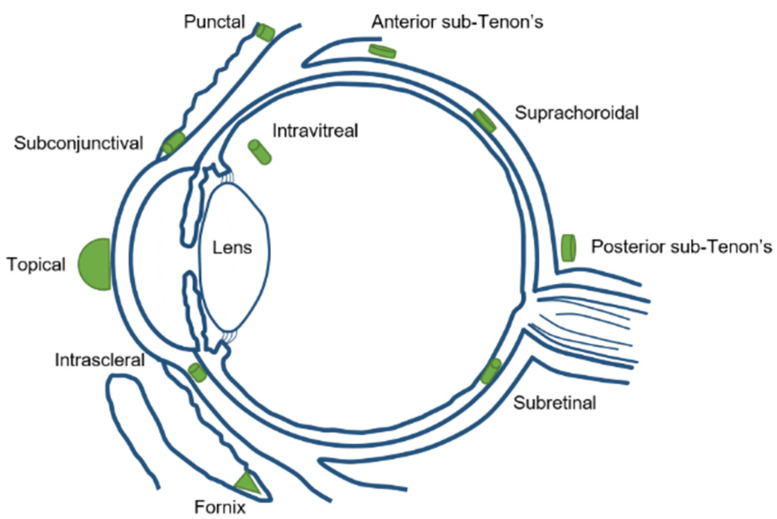
Different ocular drug delivery routes.

**Figure 3 pharmaceutics-13-01157-f003:**
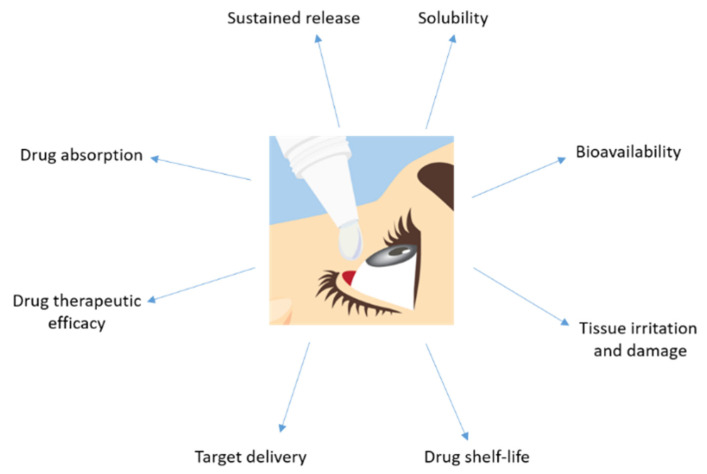
Major disadvantages showed by a drug topically administrated to the eye.

**Figure 4 pharmaceutics-13-01157-f004:**
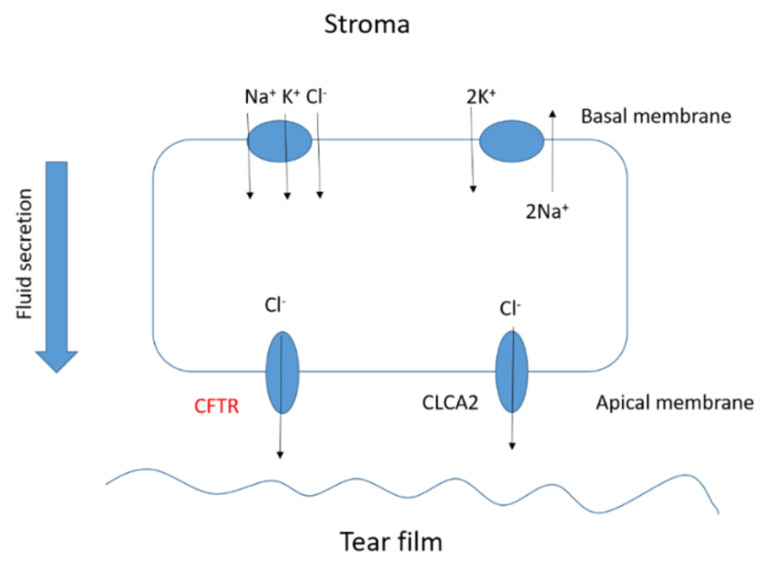
Basic scheme of the fluid transport in a corneal epithelial cell.

**Figure 5 pharmaceutics-13-01157-f005:**
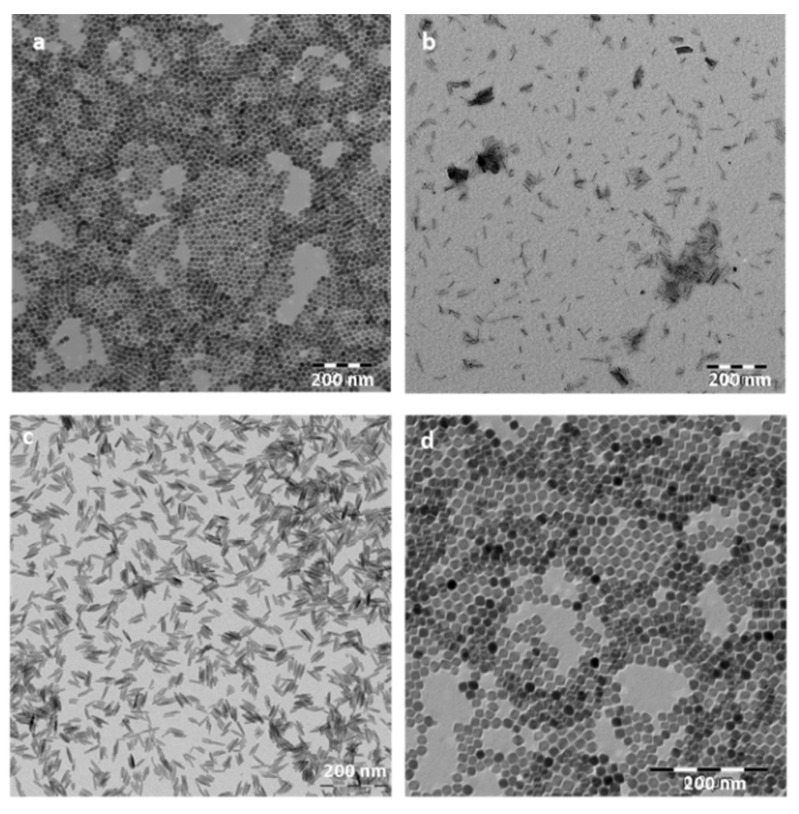
Transmission electron microscopy (TEM) images of the iron oxide nanoparticles (magnetite) prepared by the hydrothermal method with different morphologies: (**a**) spheres, (**b**) rods, (**c**) needles, and (**d**) cuboidals. Images were obtained using a LIBRA 120 Plus Carl Zeiss microscope (A Carl Zeiss SMT AG Company, Oberkochen, Germany).

**Figure 6 pharmaceutics-13-01157-f006:**
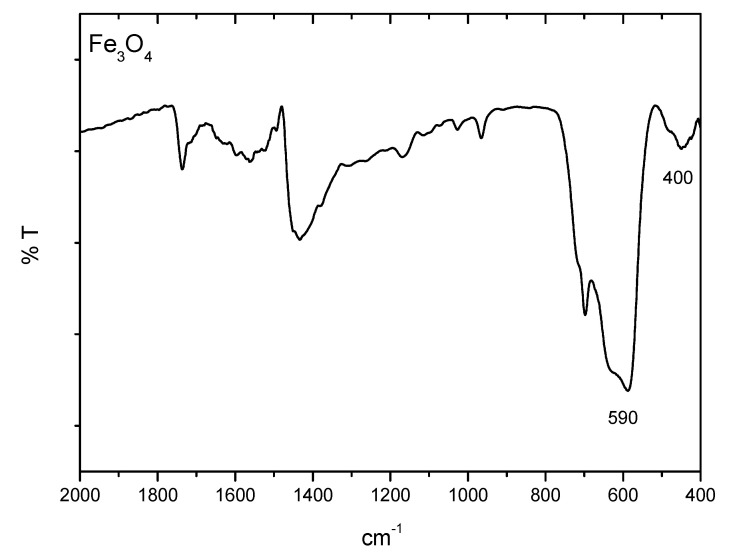
FTIR spectrum of the iron oxide nanoparticle (magnetite) spheres prepared by the hydrothermal method. This analysis was carried out in a JASCO 6200 FTIR (Japan) spectrometer with SPECTRA MANAGER V2 software. The sample was prepared by grinding 1–1.5 mg of dry particles with 150 mg of potassium bromide powder and pressing the mixture with a pellet-forming die. Analysis was done with 50 scans, at a resolution of 4 cm^−1^.

**Table 1 pharmaceutics-13-01157-t001:** Clinical nanoparticle-based strategies for ocular disease diagnostics.

Material	Size (nm)	Diagnosis Technique	Application	Reference
IONPS	10	Confocal microscopy	Retinal detachment	[83]
IONPS	50	MRI	Retinal degeneration	[84]
Ferrofluid	10–100	X-Ray Diffraction	Glaucoma treatment	[85]
Nanocubes	20	Fluorescein angiography	Glaucoma treatment	[86]
IONPS	60	Confocal microcopy	Aged Macular Degeneration (AMD), Retinal Pigment Epithelium (RPE)	[39,40]
IONPS	30	MRI	Choroidal melanoma	[87]
IONPS	200	Fluorescent confocal microscopy	Retinal degeneration	[88]

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
