# Peer review of "Advantages and Disadvantages of Using Magnetic Nanoparticles for the Treatment of Complicated Ocular Disorders"

_pharmaceutics, 2021, doi:10.3390/pharmaceutics13081157_

Round 1

Reviewer 1 Report

Dear authors:

Since, there is mention of …”magnetic nanoparticles” in the Title of manuscript,  I suggest addition of further data at list Infrared spectra of nanoparticles and its some micrographies from Scanning Electron Microscopy.

Both characterization techniques above are common in laboratory of drugs and powder characterization. The cost is very low.

Infrared spectrum can be collected from mid-infrared region, 5000 to 400 cm-1, 30-100 scans, baseline, and small smooth, with representation in the form Absorbance versus wavenumber (cm-1). The fingerprint of magnetic particle is observable, 1000 – 400 cm-1 ?

Scanning Electron Microscopy allow magnification of order of 100K times. From these micrographies shape size and agglomeration state of nanoparticles can be reached. One or more micrographies can be added to “Materials and Methods” item or in Supplementary material. But, in the body of manuscript is more interesting.

Author Response

Dear reviewer,

Thank you very much for your advice. I have made the changes that you have suggested and I think this has considerably improved  the scientific quality of this manuscript. Following your recommendations, TEM images and FTIR analysis have been added to the main body of this manuscript.

Thank you again for your comments.

Reviewer 2 Report

According to its title, this review addresses the role of magnetic nanoparticles in the treatment of cystic fibrosis-related eye disorders. However, the main focus of the manuscript ends up being ocular cancer and the role of MNPs on its management. This is quite surprising since no evidence is cited in the manuscript about the relationship between ocular cancer and cystic fibrosis. Indeed, in the section “Ocular disorders derived from Cystic Fibrosis disease”, they mention xerophthalmia, papilloedema, retinal hemorrhages, night blindness, retinal vein occlusions, blepharitis, among others, but nowhere in this section or anywhere else in the manuscript is ocular cancer related to cystic fibrosis. Therefore, before this manuscript is acceptable for publication, the authors must clarify what they really want to address in this review, ocular cancer or CF-related disorders?

Author Response

Dear reviewer,

Thank you for your comments. It is true that current uses of magnetic nanoparticles in the treatment of ocular disorders are mainly focused on the diagnosis and treatment of ocular tumors.

Our aim in this review was to extend the magnetic nanoparticles uses to other ocular disorders, specially the ocular disorders related to cystic fibrosis (CF) disease. It has been previosly demonstrated that magnetic nanoparticles are potential bacterial detection and separation agents and it considerably helps the treatment of CF patients. 

For that reason, we firstly described the ocular disorders derived from CF disease in the "Ocular disorders derived from Cystic Fibrosis disease" section and next we described the current uses of MNPs as pharmaceutical formulations. At the end of the manuscript, we have described the advantages that MNPs can offer to this kind of patients, such as improving the bioavailability of CF treatment and treating the ocular disorder with a specific target-drug, reducing the toxicity of both therapeutic agents. 

Following your recommendations, we have changed the title and we have included ocular cancer as keyword in order to make it less confusing. 

Thank you again for helping us improving the quality of our manuscript.

Sincerely,

Authors

Round 2

Reviewer 1 Report

Dear Authors:

In my oppinion the manuscript titled Advantages and Disadvantages of using magnetic nanoparticles for the treatment of Ocular disorders related Cystic Fibrosis" can be published.

Author Response

Thank you very much for the review of our manuscript.

Sincerely,

Authors

Reviewer 2 Report

  1. In the section “Ocular barriers for drug delivery systems”, a figure illustrating such barriers would make it easier to understand and appreciate the complexity of the eye and the multiple barriers that need to be overcome for efficient nanomaterials delivery.
  2. 10, line 337. The authors state the following: “The most important parameters to control in MNPs to verify its utility in diagnosis techniques are the size and shape. Both variables can affect to the dynamics of the magnetic moments (magnetic saturation, magnetic relaxation), they determine the detection capability, as well as affect to the internalization and eventual fate of MNPs 340 inside mammalians.” Although it is true that size and shape are key parameters, other aspects are also crucial in determining both the magnetic properties and the biological behavior of MNPs, such as the core crystallinity and the surface charge. These aspects should be included/commented in this section.
  3. 10, line 345: “MNPs are the most attractive nanomaterials in the treatment of different tumors, 345 including ocular tumors.” This is quite an ambitious claim. Please try to avoid categorical statements like this one unless unequivocal proof is provided. There are several throughout the article.
  4. Language editing is needed as there are several grammatical errors/inconsistencies.

Author Response

Dear Reviewer,

Thank you very much for your effort and help us improving the scientific quality of our manuscript. All your comments have been address as follows:

  • A new figure 1 has been added in the manuscript describing different ocular barriers.
  • 10, line 337. The sentence: “The most important parameters to control in MNPs to verify its utility in diagnosis techniques are the size and shape....inside mammalians.” has been modified. All the physico-chemical key parameters have been included.
  • 10, line 345: The sentence “MNPs are the most attractive nanomaterials... including ocular tumors.” has been modified in order to make it less confusing.
  • The rest of the text has been revised.

Thank you again for your help.

Sincerely,

Authors
